# Application of Genetic Resources in the Development of New Lithuanian Vegetable Cultivars

**DOI:** 10.3390/plants12040807

**Published:** 2023-02-10

**Authors:** Rasa Karklelienė, Danguolė Juškevičienė, Audrius Radzevičius

**Affiliations:** Department of Vegetable Breeding and Technology, Lithuanian Research Centre for Agriculture and Forestry, Kauno 30, Kaunas Distr., 54333 Babtai, Lithuania

**Keywords:** breeding, cultivar, genetic recourses, hybrid, productivity, vegetables

## Abstract

The investigations of genetic resources in Lithuania started in 1924. Collections including more than 110 samples of different vegetable species have been accumulated and continue to expand. The breeding program is focused on productivity, quality traits and biochemical composition, and tolerance of unfavorable environmental factors in changing climates. Currently, over 95 cultivars and hybrids of vegetables have been released, about 40 of which are included in the EU Common catalogue of varieties of vegetable species. From 2012 to 2021, twelve cultivars were developed at the Lithuanian Research Centre for Agriculture and Forestry, seven of which are fruit vegetables, four are root crops, and two are in the onion group. The breeding direction was to increase the sustainable production of vegetables and improve the quality of the products by using national and adapted genetic resources as valuable parental forms. For 10 years, vegetable plant hybrids and cultivars were developed to meet commercial cultivars’ market requirements. The tomato cultivars ‘Ainiai’ H, ‘Adas’ H, ‘Auksiai’ H, ‘Alvita’ and cucumber cultivar ‘Roliai’ are distinguished by a good biochemical composition and taste, while the sweet pepper cultivar ‘Gabija’ has high productivity and fruit quality. The carrot hybrids ‘Ieva’, ‘Rokita’, ‘Jola’, garlic cultivar ‘Dangiai’, and onion cultivar ‘Joriai’ are distinguished by their productivity and good storage ability. The Jerusalem artichoke cultivar ‘Sauliai’ has tubers with white skin and the colour of the flowers is yellow.

## 1. Introduction

Seed companies in Lithuania are recommending a wide assortment of new cultivars of vegetables. Among the large assortment of cultivars, it is not easy to choose one suitable to the growing demands for obtaining high-quality yield and the profitability of production. The growth, yield, and quality of vegetables are generally dependent on genetic, environmental, and management variables. Therefore, choosing the local cultivars or those included in the national and EU list of plant cultivars is optimal. Vegetable breeding has a long tradition in Lithuania, and it has resulted in more than 95 vegetable cultivars. Cultivars developed in Lithuania are better adapted to the local meteorological and soil conditions, and are distinguished by rich biochemical composition and taste properties. Cultivars are productive and resistant to diseases and also have an excellent market appearance [1,2,3,4,5]. When breeding vegetables, it takes on average 10–12 years to develop a new cultivar or a hybrid. Distinctiveness, uniformity, and stability (DUS) tests must be performed for the new cultivars, and their agrobiologic value is also estimated. After positive DUS test results, a new cultivar is registered in the EU Common Catalogues of Varieties of Vegetable Species and Lithuanian National List of Plant Varieties. 

Vegetables are grown worldwide in almost 200 countries and play a significant role in human nutrition [6]. Many desirable horticultural characteristics can be achieved by applying a vegetable breeding program and developing a new cultivar. In modern horticulture, new cultivars are needed that can compete with those already on the market and respond to consumers’ needs. In the development of new cultivars of vegetable plants, special attention should be paid to the European Green Course program’s goal of consuming healthy foods based on environmental sustainability. 

The most important aim in the process of plant breeding is the effective selection of crossbreeding components. Properly selected parental material and using national genetic resources well adapted to local conditions make it possible to obtain competitive cultivars. Tomato, cucumber, sweet pepper, carrot, Jerusalem artichoke, garlic, and onion are important vegetable species taking the main positions in Lithuanian and European commercial vegetable growing [7,8,9,10,11,12,13,14]. The plants of cultivars created for the last period are superior in the use of nutrients from the soil and stand out with a higher yield of marketable production [15,16,17,18,19,20,21]. Crossbreeding and directional selection are important methods playing a major role in Lithuanian vegetable breeding [22,23,24]. Individual and mass selection and synthetic mutagenic selection using ionizing gamma radiation with the biologically active chemical compound colchicine have been used to develop new valuable plants [25,26,27,28]. When hybrid breeding started, the focus was the development of new lines with exceptional properties and high combining ability. Disease-resistant tomato lines, including flower’s male sterility, were developed using these selection methods [29], and female and male flowering type lines have been developed in cucumbers as well [23]. By using back crossbreeding, an analogy of Lithuanian carrot cultivars with cytoplasmic petaloid type male sterility was created, and the combining ability of carrot CMS lines and testers on the hybrids was evaluated [30,31,32]. Homozygotic lines of onion have been evaluated and selected for future selection processes [33].

The study aims to evaluate the valuable properties of parental material of the national and adapted genetic resources of vegetables and their application in the development of new cultivars in the last decade.

## 2. Results and Discussion

The growth condition, productivity, and biodiversity of vegetable plants are affected by anthropogenic activities and climate change. Their impact on the vegetation of various ecosystems is also important and relevant in Lithuania. The collection of lines, numbers, and cultivars of national genetic resources at the SDI has been characterized by productivity and plasticity [5,25]. According to international agreements, Lithuania is committed to creating conditions for the preservation of the genetic diversity of vegetable plant species. Therefore, the possibilities for using parental material of national and adapted genetic resources in the creation of new cultivars are communicated in this article. A comprehensive assessment of the effects of abiotic and biotic factors on the productivity and qualitative parameters of cultivars will allow for the creation of the conditions for the selection and breeding of vegetable plants, and the preservation of genetic resources.

### 2.1. Fruit Vegetables

Breeding processes and research have produced significant results for the adaptability and competitiveness of vegetable cultivars under the influence of environmental and soil conditions and other factors. Cultivars created from different plant species are distinguished by their marketability. New hybrids of tomatoes and cucumbers stand out for their high yield. ‘Auksiai’ is the first Lithuanian hybrid whose fruits are orange in color. Hybrid cucumbers ‘Roliai’ are of an indeterminate species, their height reaches up to 2–3 m [21]. The new pepper cultivar ‘Gabija’ stands out for its orange fruit color [23].

#### 2.1.1. Tomato 

Over twenty years, 13 samples of tomatoes (*Solanum lycopersicum* L.) have been selected for crossbreeding. Line S09 was characterized by earliness (95 days), productivity (12 ± 2 kg m^−2^), and sugar content 4.5 ± 2%. No. 417 and No. 416 accumulated total sugar of 4.3 ± 2 and line BO-01 contains 9.0 ± 2% carotenoids. No. 1156 and lines NLT-01, SM01 are resistant to *Phytophthora infestans* (1 point) and No. 335 to *Fulvia fulva* (1 point). After evaluating the obtained progeny, seven accessions were selected as parental material with valuable traits and used to create new cultivars (Table 1). 

After testing the genetic resources in the collection, the developed tomato cultivars showed that the plants differ in productivity and fruit quality. The new hybrid of tomato ‘Auksiai’ is the first Lithuanian hybrid that has orange-colored fruits. Descriptions of the cultivars are given below.

‘Auksiai’ (BO-01XS09) is a medium-early indeterminate hybrid of tomato. The tomato hybrid author is A. Radzevičius. The first cluster forms over leaves 5–7. The fully ripened fruits are a nice orange colour and have a round shape with two or three seed sockets. The weight of the tomato fruit is about 36–40 g. The plant leaves are a dark colour. The tomatoes have a very nice and pleasant flavour and are suitable for fresh consumption and processing. The hybrid is intended for growing in greenhouses. The average yield is about 16.5 ± 1 kg m^−2^.

‘Adas’ (1156XS09) is an indeterminate, medium early hybrid of tomato. The tomato hybrid author is A. Radzevičius. The first cluster forms above the 6–7 leaves. The fruits are medium-small in size, and the average weight is approximately 30–40 g. The ripe fruit is red coloured, round with two or three seed sockets. These tomatoes have a good and pleasant taste. The average productivity reaches around 16.5 ± 1 kg m^−2^.

‘Ainiai’ (VilinaXSM01) is a determinant, medium early hybrid of tomato. The tomato hybrid author is A. Radzevičius. The first cluster forms above the 7–8 leaves. It has medium-sized fruits weighing about 70–80 g. The fruit is cylindrical-angular in shape with three or four seed sockets. It has a green spot at the base. The fruits are firm and transportable. The average productivity reaches up to 15.5 ± 1 kg m^−2^.

‘Alvita’ (BO-02XNLT-01) is a medium-late indeterminate cultivar of tomato. The tomato cultivar authors are A. Radzevičius and N. Maročkienė. The fully ripened fruits are a nice orange colour, and a slightly flat shape with four or five seed sockets. The average weight of the tomato fruit is approximately 100–120 g. The plants are resistant to fungal diseases and suitable for growing in greenhouses. The average yield is around 20.0 ± 1 kg m^−2^. 

#### 2.1.2. Cucumber 

Twelve samples were selected for the short-fruited cucumber (*Cucumis sativus* L.) plants. No 1570 was characterized by disease resistance (Sphaerotheca sp., *Erysiphe cichoracearum*—1 point), low-temperature tolerance, and ascorbic acid accumulation (12.0 ± 2 mg^−1^). No. 695, No. 1195, and No. 494 are resistant to *Erysiphe cichoracearum* (1 point). After evaluating the obtained progeny, two accessions were selected as parental material with valuable traits and used to create new cultivars (Table 2). 

The new cucumber hybrid ‘Roliai’ is an indeterminate type. The plant’s height reaches up to 2–3 m. Harvest of cucumber started respectively after 40 and 64 days from seedlings transplanting. A description of the new hybrid is given below.

‘Roliai’ (No. 1570 X No. 1758) is a parthenocarpy hybrid of cucumber. The cucumber hybrid author is E. Dambrauskas. The hybrid is intended for growing in an unheated greenhouse for a late spring-summer yield period. The yield can reach up to 15 ± 1 kg m^−2^ when cucumbers are grown in an unheated greenhouse. The cucumbers are suitable for growing in the open field, where the yield varies from 4 to 5 kg m^−2^. The fruits have an intensive green colour without a blackish shade and with light green stripes on them. The flesh of the cucumbers is distinguished by juiciness and better taste because they form thinner skin. The fruits are suitable both for fresh production and pickling. 

#### 2.1.3. Sweet Pepper 

After evaluating the sweet pepper collection and national genetic resources, seven samples were selected for a new selection. No. N-012-12 stood out with valuable characteristics and was evaluated in competitive trials. Its productivity reached 6.5 ± 0.5 kg m^−2^, its carotenoid level was 9.5 ± 2%, and it took 120 days from germination to yielding (Table 3). No. N-012-10 and cultivar ‘Alanta’ accumulated a total sugar level of 6.3 ± 2%. 

A new sweet pepper (*Capsicum annuum* L.) cultivar is distinguished by its orange fruit color and harvest starting approximately 50 days from transplanting seedlings. A description of the cultivar is given below.

‘Gabija’ (No. N-012-12) is a medium-early cultivar of sweet pepper The authors are N. Maročkienė and A. Radzevičius. The cultivar is recommended for growing in unheated greenhouses and undercover. The height of the plant reaches up to 80–90 cm and the plant habit is compact. The foliage is lush and with medium-sized leaves. The fruits are cone-shaped with pointed tips. The fruit weight is 70–85 g, the length is 11–13 cm, and the diameter is 4–5 cm. The external skin is shiny and smooth. The fruits are green-coloured before maturity, and they become orange when mature. The harvested fruits preserve a good taste for a long time. The number of fruits reaches 25–30 per plant. Total fruit yield is 6.5 ± 0.5 kg m^−2^ by planting in rows within the 70 × 40 cm distance. The average marketable yield consists of about 93%.

### 2.2. Root and Tubers Vegetables

The breeding of carrot hybrids is focused on productivity, disease resistance, and adaptation to local agroclimatic conditions. Vegetable crosses and individual or family selection were used to creating new lines and cultivars. Individual selection was used in the breeding of Jerusalem artichoke [34].

#### 2.2.1. Carrots

Carrot (*Daucus carota* subsp. *sativus* Hoffm.) Parental material was selected according to economically valuable traits from cultivars and CMS lines. Previous studies have evaluated lines that can be used to develop new cultivars. Parental material with early harvest characteristics (vegetation duration 85 ± 5 days) were identified as CMS lines NS 557 and NS 554 [9]. The parental material (CMS lines NS 557, GS 198, and NS 554) accumulated total sugar levels of 7.2 ± 1% and levels of carotene reached up to 22 ± 2 mg%. The parental form line V 316 accumulated more total sugar at 7.6 ± 1%. No. 1898 and Š 1279 were identified as the best transmitters of resistance to root and leaf diseases (1 point) (Table 4).

The process of carrots’ ontogenesis is closely related to biotic and abiotic factors. Obtaining the highest yield and the best carrot root quality is possible when all growth factors are at an optimal level. Yield is mainly affected by temperature, light, humidity, soil, and the amount of nutrients. The carrot seed germination period varies from 14 to 20 days, as confirmed by earlier investigations [16,32]. A description of the new hybrids is given below.

‘Ieva’ (ŠS 494 X No 1898) is a medium-late hybrid of Nantes-type carrots. The authors are R. Karklelienė and O. Gaučienė. The duration of vegetation is 128 ± 3 days, and fresh maturity is reached at 58 ± 3 days from sowing. The shape of the roots is cylindrical with a blunt tip, medium length (22 ± 3 cm), and diameter (4.2 ± 2 cm). The color of the phloem and xylem is an intense orange. The accumulation of carotene reaches up to 21 ± 2 mg%, the dry soluble solids are 11.5 ± 1%, and the total sugar is 7.0 ± 1%. It is suitable for growing on a flat and profiled soil surface. It is recommended to grow this hybrid in clay loam soil. The carrots are resistant to diseases, suitable for growing for the autumn harvest, and can be stored during the winter. 

‘Rokita’ (NS 557 X No 1898) is a medium-late hybrid of Nantes-type carrots. The authors are R. Karklelienė and O. Gaučienė. The duration of vegetation is 126 ± 3 days, and fresh maturity is reached at 57 ± 3 days from sowing. The shape of the roots is cylindrical with a blunt tip, medium length (23 ± 3 cm), and diameter (3.9 ± 2 cm). The color of the phloem and xylem is an intense orange. The accumulation of carotene reaches up to 22 ± 2 mg%, the dry soluble solids are 11 ± 1%, and the total sugar is 7.5 ± 1%. It is suitable for growing on a flat and profiled soil surface. It is recommended to grow this hybrid in clay loam soil. The carrots are resistant to diseases, suitable for growing for the autumn harvest, and can be stored during the winter. 

‘Jola’ (VS 39 X Š 1279) is a medium-late hybrid of Nantes-type carrots. The authors are R. Karklelienė and J. Nėniūtė. The duration of vegetation is 120–130 days, and fresh maturity is reached at 55–60 days from sowing. The shape of the roots is cylindrical with a blunt tip, medium length (23 ± 3 cm), and diameter (4.2–4.6 cm). The color of the phloem and xylem is an intense orange. The accumulation of carotene reaches up to 21 ± 2 mg%, the dry soluble solids are 11–12.5%, and the total sugar is 7.5–8.5%. It is suitable for growing on a flat and profiled soil surface. It is recommended to grow this hybrid in clay loam soil. The carrots are resistant to diseases, suitable for growing for the autumn harvest, and can be stored during the winter. 

#### 2.2.2. Jerusalem Artichoke 

According to previous studies, No. 05-1 was selected based on its valuable characteristics [34]. The expression of Jerusalem artichoke (*Helianthus tuberosus* L.) plants for tolerance to low temperatures, and good biochemical composition (5.6 ± 1% of total sugar and 4.6 ± 0.5% of ascorbic acid) was investigated in parental material (Table 5). 

The breeding process of Jerusalem artichoke started in 2005. The new cultivar was created after evaluating the expression and stability of morphological characteristics of No. 05-1. Below is the description of the new cultivar.

Jerusalem artichoke cv. ‘Sauliai’ (No. 05-1) is a perennial, herbaceous, flowering plant. The authors are N. Maročkienė, R. Karklelienė, P. Gumbelevičius. The length of the stem is 1.7–1.9 m, and the diameter is approximately 1.8–2.4 cm. The leaf is medium size, and the colour of the flowers is yellow. The size and shape of the tuber are various. The colour of the tuber skin is white. The weight of tubers reached 90–185 g, and the productivity is 86 ± 3 t ha^−1^. The tubers accumulate approximately 15.6 ± 1% of total sugar and 4.6 ± 0.5% of ascorbic acid.

### 2.3. Onion Crop Vegetables

Increasing the productivity of onion crop vegetables and preserving the quality of production is one of the most important goals in modern horticulture. Newly developed cultivars are better adapted to meteorological and soil conditions, have good biochemical composition and taste characteristics, and are productive and resistant to diseases. The development of cultivars, the study of genetic resources, and their preservation are important in the selection of onion crop plants. 

#### 2.3.1. Onion 

The old Lithuanian onion (*Allium cepa* L.) cultivars and the selection numbers selected from them are tolerant to unfavorable environmental conditions due to long-term selection and can be used for the development of new cultivars. No. 40 was characterized by tolerance to low temperature, good productivity (3.5 ± 0.3 kg m^−2^), and good biochemical composition (total sugar 7.4 ± 0.4% and ascorbic acid 12.0 ± 0.1%). No. 320 was characterized by productivity, disease resistance (*Alternaria* sp., *Puccinia alli* -1 point), and a total sugar level of 7.4 ± 0.4 ± 1% (Table 6). 

Onion ‘Joriai’ (No. 40) was characterized by tolerance to low temperatures, is well adapted to Lithuanian climate conditions and is resistant to biotic and abiotic factors. Below is a description of the new cultivar.

The onion ‘Joriai’ is a medium-early cultivar. The authors are D. Juškevičienė and R. Karklelienė. The plant vegetation lasts 90–110 days from seed germination. Onions can be multiplied by seeds or onion sets. The intensive, green-coloured foliage has 7–8 long and medium-width leaves. The cultivar forms medium-sized bulbs weighing about 90 ± 10 g. The bulb is covered with 5–6 medium-thick outer shells. The colour of the outer shell is yellowish-brown. The flesh of the bulb is white without discoloration in the epidermis. The shape of the bulb is flat round and rhombic in the longitudinal section. The onions are suitable for growing as bulbs and fresh produce and can be used for processing. The average amount of dry soluble solids reaches 14 ± 2%, and the bulbs are distinguished by very good storability.

#### 2.3.2. Garlic 

Garlic growth is disturbed when climatic conditions and growing location change. Therefore, research on their resistance to unfavorable environmental conditions is especially important. Eight local populations of hardneck garlic (*Allium sativum* L.) and introduced cultivars were studied in the collection. Phenological observations were carried out, morphological characteristics were recorded, and harvest was estimated. When determining the biological characteristics of hardneck garlic, the studied samples were evaluated according to yield, earliness (140 ± 5 days from sprouting to harvesting), and other traits. No. 24 was characterized by disease resistance, good biochemical composition (total sugar 20.0 ± 0.2% and ascorbic acid 10.0 ± 0.2%), and productivity (Table 7). No. 30 and No. 24 are resistant to *Fusarium oxysporum f.* and *Puccinia alli* (1 point).

Cultivar ‘Dangiai’ is adapted to Lithuanian climate conditions and is resistant to biotic and abiotic factors. Below is a description of the new cultivar.

Garlic ‘Dangiai’ (No. 24) is a mid-early cultivar forming foliage up to 1 m in height and a flower stem with a flower head up to 1.5 m. The authors are D. Juškevičienė and R. Karklelienė. The colour of the foliage is an intense green. Bulbils and some generative flowers form in the flower head. The size of bulbils is close to the corn grains and their amount reached up to 50 and more in a flower head. The shape of the bulb is transverse elliptic, and the colour of the external scales is white to purple with dark purple stripes. The bulb consists of 5–7 same-sized cloves that are arranged in a circle around the flower stem. The average weight of the cloves is 7 ± 3 g. The colour of the clove skin is brownish purple. The flesh is a yellowish white. The cultivar is tolerant of winter hardiness.

### 2.4. Detailed Characteristics of Newly Created Cultivars

The obtained and summarized results show that plant productivity and quality depend on the genotype and growing conditions. The marketability of all vegetable species cultivars reaches 80.2 to 97.0% (Table 8). New fruit vegetable cultivars and hybrids are productive and distinguished by fruit quality parameters. Carrot hybrids, onion, and garlic are well adapted to Lithuanian climate conditions and resistant to biotic and abiotic factors. More detailed data and features of new cultivars are submitted in the scientific articles [3,5,12,15,25].

## 3. Materials and Methods

### 3.1. Plant Material 

Thirteen cultivars were created in the period 2012–2021, seven of which were fruit vegetables, two roots, and one a tuber vegetable, and there were two cultivars to the onion group. Mass and individual selection were for developing the sweet pepper cultivar ‘Gabija’, Jerusalem artichoke cultivar ‘Sauliai’, tomato ‘Alvita’ garlic ‘Dangiai’, and onion ‘Joriai’. The new carrot hybrids ‘Ieva’, ‘Rokita’, and ‘Jola’ were developed by crossing the lines with cytoplasmic male sterility (CMS) and constant lines. Heterosis methods were applied for the development of tomato ‘Adas’, ‘Ainiai’, ‘Auksiai’, and cucumber ‘Roliai’ hybrids. Breeding material and newly created cultivars were improved according to their productivity, biochemical composition, storability, and also transportability, and resistance to biotic and abiotic factors.

### 3.2. Plant Cultivation and Methods

Investigations were carried out in the vegetable experimental field and greenhouses at the Institute of Horticulture, Lithuanian Research Centre for Agriculture and Forestry. The soil of the experimental site is a Calcic Endogleyic Luvisol (LV-gl-n-cc) light loam [35]. The vegetables were grown in a neutral soil medium rich in potassium, phosphorus, and nitrogen. The competitive investigation of new cultivars was carried out in three replications. Sowing, planting, fertilization, weeding, and plant protection were carried out according to the scientific methods for agriculture and forestry research investigations [36]. For the identification of diseases, samples were collected from damaged plants, and the visual-symptomatic diagnostics were rated according to a 0–4-point scale rate (0—no damage; 1—weak; 2—medium; 3—strong; 4—very strong) [36]. The productivity, morphological and biochemical parameters of the newly created vegetable cultivars are presented in the descriptions of the cultivars. Detailed characteristics of the newly developed cultivars for each vegetable species were evaluated by distinctiveness, uniformity, and stability [37]. The biochemical parameters (carotene, dry soluble solids, and sugar levels) were measured by fresh weight (FW) at the Laboratories of Plant Physiology and Biochemistry and Technology. Carotene was measured by the Murri method, the amount of total sugar by the Bertrane method, and dry soluble solids by a numeric refractometer. Ascorbic acid was determined using 2,6 dichlorfenolindofenol sodium chloride solution [38]. The values of the obtained data were expressed as mean ± SD (*n* = 3) (*p* < 0.05).

### 3.3. Meteorological Conditions

Weather data was collected at the Babtai agrometeorological station, using the METOS^®^sm forecasting system. The lowest air temperature in the average multi-year data of the decade during the research period was in the range 12.4–12.8 °C in May, and in September it was 13.7–14.0 °C. In the summer months (June, July, and August) the temperature varied from 15.9 °C to 20.0 °C. Based on long-term data, the amount of precipitation during the plant’s growing season averaged 39.4–78.7 mm. 

## 4. Conclusions

Parental forms of different vegetable species are characterized by high productivity and valuable qualitative features that are successfully used for breeding. The newly developed cultivars of tomato (‘Auksiai’ H, ‘Adas’ H, ‘Ainiai’ H, ‘Alvita’), cucumber (‘Roliai’ H), carrot (‘Ieva’ H, ‘Rokita’ H, ‘Jola’ H), sweet pepper (‘Gabija’), Jerusalem artichoke (‘Sauliai’), onion (‘Joriai’) and garlic (‘Dangiai’) produce high-quality marketable yields and are tolerant to environmental factors. The fruit vegetable cultivars are recommended for growing in unheated greenhouses in the spring-summer period. The cultivars of carrots and onion crop vegetables are suitable for fresh production and storage. In the changing climate conditions, these new vegetable cultivars are distinguished by productivity and quality and are competitive with cultivars grown commercially in Lithuania.

## Figures and Tables

**Table 1 plants-12-00807-t001:** Parental material from tomato national and adapted genetic resources.

Valuable Traits	Cultivar, Line, Breeding Number (No.)			
Tolerance to low temperatures (12 ± 2 °C)	‘Vilina’	‘Viltis’	‘Ryčiai’	BO-02
Productivity	No. 300	NLT-01	‘Ryčiai’	S09
Biochemical composition (Total sugar, carotenoids)	No. 417	BO-01	No. 416	S09
Earliness (95–115 days from germination to yielding)	‘Vilina’	BO-01	‘Viltis’	S09
Disease resistance (*Phytophthora infestans*, *Fulvia fulva*)	No. 335	SM01	NLT-01	No. 1156

**Table 2 plants-12-00807-t002:** Parental material from cucumber national and adapted genetic resources.

Valuable Traits	The Cultivar, Breeding Number (No.)			
Tolerance to low temperatures (12 ± 2 °C)	No. 1570	No. 1304	‘Kauniai’	‘Trakų pagerinti’
Productivity (13 ± 1 kg m^−2^)	No. 175	No. 695	No. 1925	No. 1758
Biochemical composition (Ascorbic acid)	No. 1570	No. 9	No. 1859	No. 1758
Earliness 55–60 days from germination to yielding)	No. 1295	No. 9	No. 1925	No. 1625
Disease resistance (*Sphaerotheca* sp., *Erysiphe cichoracearum*)	No. 1570	No. 695	No. 1195	No. 494

**Table 3 plants-12-00807-t003:** Parental material from sweet pepper national and adapted genetic resources.

Valuable Traits	Cultivar, Breeding Number (No.)			
Tolerance to low temperatures (12 ± 2 °C)	‘Reda’	No. 95-13		
Productivity (6.5 ± 0.5 kg m^−2^)	‘Reda’	No. N-012	‘Alanta’	No. N-012-12
Biochemical composition (total sugar, carotenoids)		No. N-012-10	‘Alanta’	No. N-012-12
Earliness (120–130 days from germination to yielding)				No. N-012-12
Disease resistance (*Botrytis cinerea*, *Alternaria* sp.)	‘Reda’	No. 027		

**Table 4 plants-12-00807-t004:** Parental material from the carrot national and adapted genetic resources.

Valuable Traits	Cultivar, Line, Breeding Number (No.)			
Tolerance to low temperature *	‘Gona’	‘Šatrija BS’	‘Garduolės’	‘Vytėnų nanto’
Productivity (7.5 ± 0.5 kg m^−2^)	ŠS 494	NS 557	VS 39	GS 888
Biochemical composition (Total sugar, carotenoids)	GS 198	NS 557	NS 554	V 316
Earliness	ŠS 494	NS 557	NS 568	ŠS 380
Disease resistance (*Alternaria* sp., *Botrytis cinerea*)	Š 1279	GS 327	ŠS 61	No. 1898

* Soil temperature 6–8 °C.

**Table 5 plants-12-00807-t005:** Parental material from Jerusalem artichoke national and adapted genetic resources.

Valuable Traits	Cultivar, Breeding Number (No.)			
Tolerance to low temperatures (−18 ± 4 °C)	No. 05-1	No. 05-3	No. 05-2	No. 05-4
Productivity (8.5 ± 0.5 kg m^−2^)	No. 05-1	No. 05-3		
Biochemical composition (Total sugar, ascorbic acid)			No. 05-2	
Earliness		No. 05-6	No. 05-5	
Disease resistance (*Phytophthora* sp.)	No. 05-1	No. 05-6		No. 05-4

**Table 6 plants-12-00807-t006:** Parental material from onion national and adapted genetic resources.

Valuable Traits	Cultivar, Breeding Number (No.)			
Tolerance to low temperatures *		No.40	No.326	
Productivity (3.1 ± 0.3 kg m^−2^)	‘Babtų didieji’	No.40	No.326	No.320
Biochemical composition (Total sugar, ascorbic acid)		No.40		No.320
Earliness (110 ± 10 days from germination to yielding)	‘Babtų didieji’	No.80	No.157	
Disease resistance (*Alternaria* sp., *Puccinia alli)*	No.320	No.326	No.326	No.320

* Soil temperature 6–8 °C.

**Table 7 plants-12-00807-t007:** Parental material from garlic national and adapted genetic resources.

Valuable Traits	Cultivar, Breeding Number (No.)			
Tolerance to low temperature (−20 ± 5 °C)	‘Žiemiai’	No.5		
Productivity (2.1 ± 0.3 kg m^−2^)	‘Žiemiai’	No.5	No.4	No. 24
Biochemical composition (Total sugar, ascorbic acid)			No.4	No. 24
Earliness	No.30	No.6	No.7	
Disease resistance (*Fusarium oxysporum f.*, *Puccinia alli)*	No.30	No.30		No.24

Valuable Traits

**Table 8 plants-12-00807-t008:** Agrobiological features of newly created vegetable cultivars.

Features	Vegetable Crops and Respective Cultivars											
	Tomato				Cucumber	Sweet Pepper	Carrot			Jerusalem Artichoke	Onion	Garlic
	‘Ainiai’ H	‘Adas’ H	‘Auksiai’ H	‘Alvita’	‘Roliai’ H	‘Gabija’	‘Ieva’ H	‘Rokita’ H	‘Jola’ H	‘Sauliai’	‘Joriai’	‘Dangiai’
Plant height (m)	1.5 ± 0.6	2.4 ± 0.7	2.4 ± 0.7	2.5 ± 1.0	1.6 ± 1.0	0.85 ± 1	***	***	***	1.6 ± 1	0.8 ± 0.1	1.4 ± 0.4
Leaf colour intensity, point *	7	5	5	5	5	5	5–6	6	5	3	7	7
Fruit, root, bulb outer skin colour intensity, point *	5	5	3	5	3	5	5–6	6	5–6	1	1	2
Fruit, root, bulb length (cm)	***	***	***	***	8 ± 1	12 ± 1	22 ± 3	23 ± 3	23 ± 3	***	***	***
Productivity (kg m^−2^)	15.5 ± 1	16.5 ± 1	16.5 ± 1	20.0 ± 1	15.0 ± 1	6.5 ± 0.5	8.5 ± 1	7.2 ± 1	7.8 ± 1	8.6 ± 1	3. ± 0.1	2.2 ± 0.5
Marketability (%)	92.0	93	92.9	90.0	89.6	92.8	95.0	93.0	92.0	80.2	94.0	97.0
Duration of vegetation (days)	105 ± 2	113 ± 2	113 ± 2	107 ± 2	92.0 ± 1	121 ± 1	128 ± 3	126 ± 3	125 ± 5	***	108 ± 5	143 ****

* Estimation according to UPOV guidelines; *** not observated; **** days from sprouting.

## Data Availability

Not applicable.

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
