# Peer review of "Application of Genetic Resources in the Development of New Lithuanian Vegetable Cultivars"

_plants, 2023, doi:10.3390/plants12040807_

Round 1

Reviewer 1 Report

The manuscript provides an interesting summary of the use of vegetable germplasm in breeding and its results. It is easy and engaging to read and offers wide information.

The article is designed as an Article, but the content is more in line with a Review-type publication. Not an objection, but maybe the authors can think about turning it into a Review article?

In order to further improve the quality of the manuscript, I would have a few more suggestions for minor corrections:
- presentation of tables - the manuscript includes several small tables in which I cannot really understand the meaning of bold and italic usage.
- the text uses the terms 'cultivar' and 'variety' with the same meaning - one of them should be used.

Author Response

Please see the attachment for answers to Reviewer 1

Reviewer 2 Report

The paper presents some critical aspect of which, the most important one is the missing of a discussion of the results. In this regard I think that should be improved with a dedicated “Discussion” paragraph or inserted in the “Results” section.

Other specific comments:

-        Please replace the list of citations “[1,2,3,4,5]” as “[1-5]” in line 37 and throughout the text.

-        Line 41: “and” replace with “and”.

-        Line 169: the Latin name of a species should be in italics.

-        In my opinion the lines 294-300 are not competent in the Material and Method section.

-        Please add a space “ °C” before the degree symbol in lines 327-328.

Author Response

Please see to attachment answer to reviewer 2

Round 2

Reviewer 2 Report

Point 1: I asked to improve the manuscript with a discussion section in which the results are discussed with a comparison with the actual literature, but the authors added “Discussion” in the results section without any effort to improve this section. In addition, the tables (or supplementary tables) with statistical analysis for each valuable traits in each cultivar should be inserted.

Point 2: I asked to replace the list of citations “[1,2,3,4,5]” as “[1-5]” in line 37 and THROUGHOUT the text, but only in the line 37 was corrected.

Point 3: “Daucus sativus” (line 172 of the present version, line 169 of the old version) is not an English name.

Point 4: Is the material arranged in a randomized block design with three replications? It’s not clear in the experimental design. Please specify.

Author Response

In response to the reviewer's comments, please see the attachment.

Round 3

Reviewer 2 Report

I suggest this revised version of manuscript for publication.

Author Response

Thank you very much.
